# Investigation of Sperm and Seminal Plasma Candidate MicroRNAs of Bulls with Differing Fertility and In Silico Prediction of miRNA-mRNA Interaction Network of Reproductive Function

**DOI:** 10.3390/ani12182360

**Published:** 2022-09-09

**Authors:** Vanmathy Kasimanickam, Nishant Kumar, Ramanathan Kasimanickam

**Affiliations:** 1Center for Reproductive Biology, College of Veterinary Medicine, Washington State University, Pullman, WA 99164, USA; 2National Dairy Research Institute, Indian Council of Agricultural Research, Karnal 132001, India; 3Department of Veterinary Clinical Sciences, College of Veterinary Medicine, Washington State University, Pullman, WA 99164, USA

**Keywords:** dairy bulls, fertility, sperm, seminal plasma, micro RNA, mRNA, bioinformatics, reproductive function

## Abstract

**Simple Summary:**

The objective of this study was to identify differentially expressed (DE) sperm and seminal plasma microRNAs (miRNAs) in high- and low-fertile Holstein bulls (four bulls per group), integrate miRNAs to their target genes, and categorize target genes based on predicted biological processes. Out of 84 bovine-specific, prioritized miRNAs analyzed by RT-PCR, 30 were differentially expressed in high-fertile sperm and seminal plasma compared to low-fertile sperm and seminal plasma, respectively (*p* ≤ 0.05, fold regulation ≥5 magnitudes). Interestingly, expression levels of DE-miRNAs in sperm and seminal plasma followed a similar pattern. Highly scored integrated genes of DE-miRNAs predicted various biological and molecular functions, cellular process, and pathways. Further in silico analysis revealed categorized genes may have a plausible association with pathways regulating sperm structure and function, fertilization, and embryo and placental development. In conclusion, highly DE-miRNAs in bovine sperm and seminal plasma could be used as a tool for predicting reproductive functions. Since the identified miRNA-mRNA interactions were mostly based on predictions from public databases, the causal regulations of miRNA-mRNA and the underlying mechanisms require further functional characterization in future studies.

**Abstract:**

Recent advances in high-throughput in silico techniques portray experimental data as exemplified biological networks and help us understand the role of individual proteins, interactions, and their biological functions. The objective of this study was to identify differentially expressed (DE) sperm and seminal plasma microRNAs (miRNAs) in high- and low-fertile Holstein bulls (four bulls per group), integrate miRNAs to their target genes, and categorize the target genes based on biological process predictions. Out of 84 bovine-specific, prioritized miRNAs analyzed by RT-PCR, 30 were differentially expressed in high-fertile sperm and seminal plasma compared to low-fertile sperm and seminal plasma, respectively (*p* ≤ 0.05, fold regulation ≥ 5 magnitudes). The expression levels of DE-miRNAs in sperm and seminal plasma followed a similar pattern. Highly scored integrated genes of DE-miRNAs predicted various biological and molecular functions, cellular process, and pathways. Further, analysis of the categorized genes showed association with pathways regulating sperm structure and function, fertilization, and embryo and placental development. In conclusion, highly DE-miRNAs in bovine sperm and seminal plasma could be used as a tool for predicting reproductive functions. Since the identified miRNA-mRNA interactions were mostly based on predictions from public databases, the causal regulations of miRNA-mRNA and the underlying mechanisms require further functional characterization in future studies.

## 1. Introduction

MicroRNAs (miRNAs) are small non-coding RNAs, expressed in various tissues [1,2,3] and in biofluids [3,4,5,6] that regulate the target genes’ expression post-transcriptionally and translationally. Extracellular miRNAs are detected in various body fluids, including blood serum and plasma, urine, saliva, semen, and milk [7,8]. Although miRNAs are potential biomarkers for developmental stages and diseases, neither the origin of extracellular miRNAs nor their communication between cells of origin and the target cells, are well known. Such studies are hindered by the challenges of isolating and characterizing RNA. Recently, we characterized mature miRNA expression patterns in boar sperm and seminal plasma [9,10].

Seminal plasma is a multifaceted biofluid that provides a nutritious and protein-rich environment that is necessary for the proper functioning of sperm such as sperm maturation, metabolism, motility, modification of sperm membranes, capacitation, acrosome reaction, interaction with oviductal and uterine epithelium, and fertilization [11,12]. Components of seminal plasma are energy substrates (fructose, sorbitol, pyruvate, and glyceryl phosphocholine), organic compounds (citric acid, peptides, proteins, lipid, hormones, and cytokines) including essential (threonine, tryptophan, and lysine) and non-essential amino acids (arginine, glycine, serine, tyrosine, and phenylalanine), nitrogenous compounds (ammonia, urea, uric acid, and creatinine), ions (Na^+^, K^+^, Zn^2+^, Ca^2+^, Mg^2+^, Cl^−^, and PO_4_^3−^), reducing substances (ascorbic acids and hypo-taurine), and reactive oxygen species scavenging enzymes (glutathione peroxidase 5, thioredoxin, glutathione-s-transferase M1, superoxide dismutase 1, and peroxiredoxin) [13,14,15,16].

MicroRNAs are highly stable in biofluids. Expression of miRNAs is detectable in sperm and seminal plasma of all animal species [17,18,19,20,21,22]. The miRNA expression is greater in sperm compared to seminal plasma and follows a similar expression trend [9,10]. The miRNAs may transmigrate between sperm and seminal plasma [23,24]. Bovine sperm and seminal plasma consist of or are associated with several miRNAs; however, functions of most miRNAs remain unknown, and several miRNAs are involved in fertilization [25,26]. Absence, presence, under-, or over-expression of specific genes regulated by miRNAs could alter sperm functions, reduce fertilizing ability, and thus lower the fertility of an ejaculate. The objective of this study was to elucidate differentially expressed (DE) miRNAs in sperm and seminal plasma of high- and-low fertile bulls, integrate miRNAs to their target genes, and categorize target genes to predict biological processes. The hypothesis was that sperm and seminal plasma miRNAs expression differ between high- and low-fertile bulls. Differentially expressed miRNAs of high-fertile bulls and their top-ranked integrated target genes will be involved in the biological process critical for regulating sperm function, fertilization, and embryo development.

## 2. Materials and Methods

### 2.1. Ethics Statement

This study was performed in strict accordance with the ethics, standard operating procedure, and use of the animal cells and biofluids for research. The protocol was approved by the institutional animal care and use committee of the Washington State University.

### 2.2. Semen Sample Processing

Fresh bull semen was collected during the spring season using an artificial vagina. Holstein bulls (*n* = 8; age 3.4 ± 0.11 year) with high (*n* = 4) and low (*n* = 4) fertility based on sire conception rates (https://www.aipl.arsusda.gov/reference/arr-scr1.htm and https://uscdcb.com) (accessed on 10 August and 7 December 2016) were selected [Appendix A; high-fertility bulls with SCR ≥ 4 (mean reliability: 91%); low fertility bulls with SCR ≤ −2 (mean reliability: 82%), 6 percentage points mean difference in sire conception rates between high vs. low fertile bulls in this study]. Sire conception rate (SCR) evaluation is released for active artificial insemination Holstein bulls that have a minimum of 300 breeding in the last 48 months (at least 100 breeding in the last 12 months) in at least 10 herds. Immediately after the collection, % progressive motility and % abnormal spermatozoa were determined using Society for Theriogenology standards. None of the semen samples contained either immature germ or somatic cells. Undiluted semen was centrifuged at 1000× *g* at 4 °C for 20 min. The sperm pellet was washed in PBS and centrifuged three times to ensure the removal of seminal plasma. Thereafter, the seminal plasma supernatant was centrifuged at 16,000× *g* at 4 °C to ensure the removal of residual sperm. Aliquots of the sperm pellets and seminal plasma supernatant were flash-frozen to −196 °C and stored at −80 °C. Prior to use, it was thawed at room temperature for 15 min. All of the procedures were performed for sperm and seminal plasma samples from each bull separately.

#### 2.2.1. Individual Progressive Motility (%)

Individual progressive motility was assessed in an ejaculate diluted with warmed semen extender. A drop of diluted sperm was placed on a slide, covered with a coverslip, and examined at 400× using phase contrast microscopy. The proportion of sperm that were moving progressively across the field of view was estimated.

#### 2.2.2. Abnormal Sperm (%)

The morphology of individual sperm was determined by examining an eosin-nigrosin stained semen smear under oil immersion (1000×). A 4 or 5 mm drop of warm stain near the end of a warm microscopic slide and a drop of semen near the stain were placed, mixed using a Pasteur pipette, and a smear was prepared by pulling a drop of stained semen slowly across the slide. Percentage abnormal sperm was determined by estimating morphology of at least 100 sperm.

### 2.3. RNA Isolation

#### 2.3.1. Sperm

RNA was isolated from sperm [9,10] using RNeasy plus Universal Mini Kit (Qiagen Inc., Valencia, CA, USA), following the manufacturer’s instructions. In brief, 750 μL QIAzol Lysis Reagent (Qiagen Sciences Inc., Germantown, MD, USA) was added to the sperm pellet (approximately 100 × 10^6^ sperm), completely homogenized, and held at room temperature for 5 min to help dissociation of nucleoprotein complexes. Then, 100 μL genomic DNA (gDNA) eliminator solution was added, the mixture was shaken vigorously to eliminate gDNA, and then 150 μL of chloroform was added. Following vigorous shaking and 2 to 3 min incubation at room temperature, the mixture was centrifuged (12,000× *g*, 15 min, 4 °C), the upper aqueous phase was harvested, 1.5 volume of 100% ethanol was added, and the mixture was mixed thoroughly by vigorous pipetting. The sample was then layered on a RNeasy mini spin column (included in the RNeasy Plus Universal Mini Kit), centrifuged (≥8000× *g*, 15 s, room temperature) and bound RNA was washed by centrifugation (≥8000× *g*, 15 s, room temperature) using buffers RWT and RPE consecutively. The RNA then was eluted using 60 × L RNase-free water. The concentration was measured in a Thermo Scientific NanoDrop 1000 Spectrophotometer (Thermo Fisher Scientific Inc., Waltham, MA, USA). The purity of RNA was determined by determining the ratio of absorbance at 260 and 280 nm and samples were stored at −80 °C.

#### 2.3.2. Seminal Plasma

Small RNAs were purified from seminal plasma [9,10] using a miRNeasy serum/plasma kit (Qiagen Inc., Valencia, CA, USA). The kit included phenol/guanidine-based lysis of samples and silica-membrane column-based isolation of small RNAs. The kit was designed to isolate cell-free small RNAs. QIAzol reagent (750 μL) was added to 150 μL of thawed samples, mixed thoroughly, incubated for 5 min at room temperature, and then 3.5 μL miRNeasy serum/plasma spike-in-controls (lyophilized C. elegans miR-39 mimic; 1.6 × 10^8^ copies/μL) and 150 μL chloroform were added. The mixture was shaken vigorously for 15 s, incubated for 2 to 3 min at room temperature, and then centrifuged (12,000× *g* for 15 min at 4 °C). The upper aqueous phase was harvested, avoiding the interphase, mixed with ~1.5 volumes of 100% ethanol and mixed by pipetting. Approximately half of the mix was transferred into a RNeasy MinElute spin column in a 2 mL collection tube and centrifuged (12,000× *g*, 15 s, room temperature). The flow-through was discarded and this step was repeated with the rest of the sample using the same column. The RNA was bound to the membrane; this membrane with bound RNAs was washed sequentially through 700 μL of buffer RWT and 500 μL of buffer RPE (≥8000× *g*, 15 s, room temperature). Then, the bound RNA was washed through 500 μL of 80% ethanol by centrifugation (≥8000× *g*, 2 min, room temperature). The RNeasy MinElute column was dried by centrifugation (12,000× *g*, 5 min), and the flow-through and collection tube were discarded. Small RNAs were then eluted by placing the membrane in 24 μL RNase-free water and centrifuging (12,000× *g*, 1 min).

### 2.4. Complementary DNA Synthesis of Sperm and Seminal Plasma miRNAs

Total RNA containing miRNA was used as starting material. Mature miRNA was reverse transcribed into cDNA using miScript II RT kit (Qiagen Inc., Valencia, CA, USA). In brief, template RNA was thawed on ice, and 10× miScript Nucleics mix, 5× miScript HiSpec buffer, and RNase-free water were thawed at room temperature. Reaction components for a 20 μL reaction were 4 μL of HiSpec buffer, 2 μL Nucleics mix, 2 μL of reverse transcriptase enzyme mix, and 12 μL of RNA template containing 250 ng RNA. Reverse-transcription reaction components were gently mixed, briefly centrifuged (2000× *g*, 10 s), and kept on ice. The mixture was incubated at 37 °C for 60 min and then at 95 °C for 5 min in a Thermocycler (Thermo Fisher Scientific, San Francisco, CA, USA). After incubation, the product was placed on ice. The product containing cDNA equivalent to 250 ng RNA was diluted with 90 μL nuclease-free water and stored at −20 °C prior to real-time PCR.

### 2.5. Sperm and Seminal Plasma Mature Mirna Profiling Using Real-Time PCR

Real-time PCR profiling of sperm and seminal plasma mature miRNAs using miScript miRNA PCR arrays in combination with the miScript SYBR Green PCR Kit (Qiagen Sciences Inc., Germantown, MD, USA), which contained the miScript Universal reverse primer and QuantiTect SYBR Green PCR Master Mix, was performed. A bovine miRNome miScript miRNA PCR array 96-well plate 1 (Table 1) was used in this study. The array plate consisted of specific primers to identify 84 highly prioritized bovine mature miRNAs from the most current miRNA genome database, as annotated in miRBase version 20 (www.miRBase.org) (accessed on 19 August 2015) [5]. A set of controls facilitated threshold cycle normalization, assessing reverse transcription performance, and assessing PCR performance. It should be noted that the 84 miRNAs specific to bovine species were selected based on the information available via experiments and computations in the literature and miRbase. In addition, genes involved in male reproductive functions and associated miRNAs were also taken into consideration.

The reaction mix was prepared with 1375 μL of 2× QuantiTect SYBR Green PCR master mix, 275 μL of 10× miScript universal primer, 1000 μL of RNase free water, and 100 μL of template cDNA for each 96-well plate. A 25 μL volume of the reaction mixture was added to each well and the template was amplified in StepOnePlus cycler (Applied Biosystems, Foster City, CA, USA). Cycling conditions consisted of an initial heating step at 95 °C for 15 min. Forty cycles included 15 s denaturation step at 94 °C, 30 s annealing step at 55 °C, and 30 s extension step at 70 °C. Dissociation curve analysis was performed to verify specificity and identity.

### 2.6. Bovine Mature miRNA PCR Array Analysis

Eighty-four high-priority bovine mature miRNAs were selected from the miRNA genome database to elucidate DE-miRNAs in sperm and seminal plasma samples. Reverse transcription and positive controls were chosen to ensure the efficiency of the array, reagents, and instrument. Raw CT data (.xlsx file format) were uploaded to the data analysis center (http://pcrdataanalysis.sabiosciences.com/pcr/arrayanalysis.php) (accessed on 10 May 2017 and 16 July 2018). Data quality control was examined to assess amplification reproducibility and reverse transcription efficiency, and to detect any other contamination in amplified samples [5,10]. The controls were cel-miR-39-3p, SNORD61, SNORD68, SNORD72, SNORD95, SNORD96A, RNU6-2, miRTC, and PPC. The CT values of samples were calibrated to the CT values of cel-miR-39-3p. Global CT mean of expressed miRNAs was chosen to normalize the target sperm and seminal plasma miRNAs. SNORDs and RNU6-2 served as internal normalizers. Two reverse transcription controls and two positive controls ensured the efficiency of the array, the reagents, and the instrument. The distribution of CT values and raw data average in both groups were reviewed. Average ΔCT, 2^−ΔCT^, fold change, P-value, and fold regulation were calculated in the web-based program, and *p*-values were included in subsequent graphical analyses. Average CT values were converted into linear 2^−ΔCT^ values and *p*-values were calculated with a student’s *t*-test.

### 2.7. Bioinformatics Analysis

#### 2.7.1. Conserved Nucleotide Sequences

Nucleotide sequences of DE-miRNAs of bovine, human, and mice were retrieved from miRBase (www.mirbase.org) (accessed on 10 December 2021). Sequences of miRNAs from humans and mice were compared with the sequences of bovine species to ensure the similarity [27,28].

#### 2.7.2. Identification of Target and Predicted Genes of Differentially Expressed miRNAs

The target genes of DE-miRNAs were predicted using miRNet (http://www.mirnet.ca/) (accessed on 16 December 2021) [29]. This tool integrates data from different miRNA databases (TarBase, miRTarBase, and miRecords). The prediction analysis was performed for upregulated and downregulated DE-miRNAs separately.

#### 2.7.3. Construction of Protein-Protein Interaction Network and Screening of Hub Genes

The protein-protein interaction (PPI) network of DE-miRNAs’ predicted target genes was performed by the Search Tool for the Retrieval of Interacting Genes/Proteins (STRING) online database (http://stringdb.org/) (accessed on 16 December 2021) [30]. Gene Ontology (GO) functional annotation for biological process and Kyoto Encyclopedia of Genes and Genomes (KEGG) pathway enrichment analysis for predicted target genes of DE-miRNAs were performed. A *p*-value of <0.05 was regarded as statistically significant. The PPI interaction was exported to the Cytoscape software (version 3.9) and visualized [31]. The hub genes were selected out as the top 20 nodes of the PPI network using the Maximal Clique Centrality (MCC) method [32], which had a better performance on the precision of predicting top essential proteins. Further analysis was performed using ClueGO (Robertson and Sharkey, 2016) to integrate GO terms as well as KEGG pathways and create a functionally nested or organized GO/pathway term (k score = 3). This task analyzes one set of genes or compares two lists of genes and comprehensively visualizes functionally grouped genes [33].

#### 2.7.4. Gene Ontology and Functional Annotation Analysis

To understand the biological meaning behind DE-miRNAs and integrated genes, biological processes were investigated in the PANTHER (Protein ANalysis THrough Evolutionary Relationships) Classification System. (www.geneontology.org/, www.pantherdb.org/) (accessed on 22 December 2021). Further, the important GO terms and their pertinent cooccurring terms were analyzed using QuickGO (https://www.ebi.ac.uk/QuickGO) (accessed on 22 December 2021).

#### 2.7.5. Real-Time Polymerase Chain Reaction for Determining mRNA Expression of Hub Genes

Genes such as dna methyltransferase 1 (*DNMT1*), forkhead box P3 (scurfin) (*FOXP3*), phosphatase and tensin homolog (*PTEN*) and Zinc Finger E-Box Binding Homeobox 1 (*ZEB1*) were selected from the group of hub genes to substantiate the mRNA expressions in sperm and seminal plasma samples.

Total RNA extraction and complementary DNA synthesis were performed as previously described [34,35]. In brief, sperm and seminal plasma samples were used to extract RNA by TRizol (Invitrogen, Carlsbad, CA, USA). The RNA concentration and quality were determined using NanoDrop 1000 spectrophotometer and all RNA samples were treated with DNAse I (Invitrogen) to remove the DNA contaminant. Complementary DNA was synthesized using the iScript cDNA synthesis kit (Bio-Rad Laboratories Inc., Hercules, CA, USA) and stored at −20 °C.

Specific primer pairs (Table 2) for the hub genes were designed using primer-BLAST (www.ncbi.nlm.nih.gov/tools/primer-blast/, accessed on 1 July 2022). Prior to real-time PCR, ethidium bromide-stained electrophoresis gel for the amplicon of the expected size was performed (Appendix A). Real-time PCR was conducted using Fast SYBR Green Master Mix (Applied Biosystems, Foster City, CA, USA) as previously described [25] following the manufacturer’s instruction. Endogenous control glyceraldehyde-3-phosphate dehydrogenase (GADPH) was used to normalize the threshold cycle (CT) values. Fold comparisons were made between the high- and low-fertile groups.

#### 2.7.6. Protein Immunoblots

Proteins such as DNMT1, FOXP3, PTEN, and ZEB1, were selected to substantiate the protein expressions in sperm and seminal plasma samples.

Western blots for sperm and seminal plasma samples were performed by methods described previously [36]. In brief, protein extraction methods included: addition of protease and phosphatase inhibitor to sperm and seminal plasma samples accordingly; homogenization; lysate incubation at 4 °C for 45 min; centrifugation (at 12,000× *g* for 20 min); and determination of protein concentrations. Protein samples were then electrophoresed through 12% SDS-PAGE gel (Bio-Rad Laboratories, Philadelphia, PA, USA) and then transferred onto PVDF membrane (Bio-Rad Laboratories, Hercules, CA, USA). Sixty micrograms of protein lysate were used per lane. The membrane blots were incubated in 10% goat serum in PBS to block non-specific binding. After overnight incubation at 4 °C with primary antibodies (mouse monoclonal to DNMT1 (Catalog # MA5-16169), rabbit polyclonal to PTEN (Catalog # 600-401-859), and mouse polyclonal to ACBT (sc-47778) from Santa Cruz Biotechnology, Santa Cruz, CA, USA), membranes were washed in buffer containing 2% animal serum and 0.1% detergent. The membranes were then incubated in secondary antibodies (goat anti-mouse IgG-FITC for DNMTA1 and ACBT (sc-2010; Santa Cruz Biotechnology, Dallas, TX, USA) and goat anti-rabbit IgG-FITC for PTEN (sc-2012; Santa Cruz Biotechnology, Dallas, TX, USA) for 1 h at room temperature. The blots were then washed and scanned using the Pharos FX Plus system (Bio-Rad Laboratories, Hercules, CA, USA). FITC fluorophore was excited at 488 nm and read at the emission wavelength of 530 nm. All possible negative controls were included.

#### 2.7.7. Statistical Analyses to Determine Differences in mRNA Expression

The average mRNA CT values were converted into linear 2^−ΔCT^ values and differences in relative expressions between high- and low-fertile bulls were calculated with a student’s *t*-test. Data were analyzed with a statistical software program (SAS version 9.4 for Windows, SAS Institute, Cary, NC, USA). Correlation coefficients were estimated using PROC CORR to determine the association between miRNA and mRNA fold changes. The RT-PCR data were analyzed by *t*-test, using 2-DDCt values to ascertain statistical significance of any differences in mRNA expressions between high- and low-fertile bulls. For all statistical analyses, *p* ≤ 0.05 was considered significant.

## 3. Results

The sire conception rate (SCR), progressive motility (%), and abnormal sperm (%) for high- and low-fertile Holstein bulls used in the study are given in Appendix A.

On miRNA semiquantitative profiling, 32 miRNAs were differentially expressed (*p* ≤ 0.05; fold change magnitude cut-off at 5, high) in high compared with low-fertile sperm and seminal plasma (Figure 1A). Of these, 20 miRNAs were upregulated (≥5), and 12 miRNAs were down regulated (≤−5) in the high-fertile sperm and seminal plasma relative to low-fertile sperm and seminal plasma (*p* < 0.05). When fold change cut-off at 2 was considered, there were 56 miRNAs (33 upregulated (≥2) and 23 down regulated (≤−2)) were differentially expressed in high compared with low-fertile sperm and seminal plasma (*p* ≤ 0.05). Further, when fold change cut-off at 10 (very high) was considered, there were 11 miRNAs (9 upregulated (≥10) and 2 down regulated (≤−10)) were differentially expressed in high compared with low-fertile sperm (*p* < 0.001); whereas 10 miRNAs (6 upregulated (≥10) and 4 down regulated (≤−10)) were differentially expressed in high compared with low-fertile seminal plasma (Figure 1B; *p* < 0.001).

Interestingly, the differential expression of miRNAs in high- vs. low-fertile sperm and seminal plasma followed a similar trend. Fold changes were slightly higher for all upregulated and slightly lower for all downregulated miRNAs in sperm compared with seminal plasma. Fold changes for all 84 miRNAs in sperm and seminal plasma for high-fertile bulls (relative to low-fertile bulls) are given in Appendix A, respectively. The ratio of sperm: seminal plasma miRNA fold expression differences varied from 0.75 to 1.40 in high-fertile bulls. Since DE-miRNAs (upregulated and downregulated) had a similar pattern for sperm and seminal plasma, the miRNA-mRNA interaction analysis, construction of PPI network, and Gene Ontology and functional annotation analysis were performed once using differentially expressed miRNAs with a fold change cut-off at 5.

Nucleotide sequence similarities for the DE-miRNAs for humans, mice, and cattle are presented in Appendix A. Bovine sequences were very similar to human and mouse nucleotide sequences. Therefore, human miRNA IDs were used to construct miRNA-mRNA interaction network and functional enrichment analysis. 

The miRNA-mRNA interaction analysis for the upregulated miRNAs resulted in 27881 target genes and 75 predicted genes (Appendix A). The PPI for the 75 predicted genes (73 nodes and 188 edges, PPI enrichment *p* < 1.0 × 10^−16^ revealed 299 significantly enriched GO terms biological processes (False Recovery Rate, *p* < 0.05) and 35 significant (False Recovery Rate, *p* < 0.05) KEGG enrichment pathways (Appendix A). The miRNA-mRNA interaction analysis for the downregulated miRNAs resulted in 24380 target genes and 57 predicted genes (Appendix A). The PPI for the 57 predicted genes (56 nodes and 186 edges, PPI enrichment *p* < 1.0 × 10^−16^) revealed 391 significantly enriched GO terms biological processes (False Recovery Rate, *p* < 0.05) and 50 significant (False Recovery Rate *p* < 0.05) KEGG enrichment pathways (Appendix A).

The PPI networks for predicted genes for upregulated miRNAs and downregulated miRNAs were separately constructed (Figure 2A,B) using the STRING database and the Cytoscape software. According to the degree (MCC method), the top 20 hub genes in the networks for upregulated miRNAs and downregulated miRNAs were screened and presented in Figure 3A,B. To decipher functionally nested gene ontology and pathway annotation networks for the predicted genes of upregulated and downregulated miRNAs in higher fertile sperm and seminal plasma, ClueGo nested network analysis was performed, and the results are presented in Figure 4 and Figure 5.

Differentially expressed miRNAs, associated hub genes, and their linked reproductive functions are given in Table 3. Finally, GO terms such as fertilization, implantation, trophoblast formation, placental development, and in utero embryonic development (from the 20 hub gene predicted 640 terms) were used to retrieve (QuickGO) relevant co-occurring GO terms (Appendix A) and from the GO functional annotation terms, fertilization (Figure 6A) and progeny development (Figure 6B) were constructed.

The mRNA expressions for ZB1 and DNMT1 were greater (*p* < 0.05) in high compared to low-fertile bulls (Figure 7); whereas the mRNA expressions for PTEN and FOXP3 were lower in high compared to low-fertile bulls (*p* < 0.05; Figure 6). Protein immunoblots were performed to recognize the presence of PTEN, DNMT1, and ACBT (reference gene) proteins in sperm and seminal plasma. The PTEN, DNMT1, and ACBT proteins were 47, ~180, and 42 kDa, respectively (Appendix A). The miRNA relative fold change and associated mRNA relative fold change (miRNA- mRNA pair) showed a negative correlation (Figure 8, *r* = −0.86; *p* < 0.05).

## 4. Discussion

The goal of this current investigation was to elucidate differentially expressed miRNAs in high-fertile sperm and seminal plasma compared to low-fertile sperm and seminal plasma, and to use bioinformatics to investigate how top-ranked integrated genes of DE-miRNAs exerts wide-ranging connotations in sperm function, fertilization, and progeny development. The current study investigated prioritized miRNAs in bull sperm and seminal plasma using real-time PCR. Real-time PCR profiling employed in this study eliminated the need for validation experiments associated with microarray technology. The result showed that 20 miRNAs were highly (≥5 fold) upregulated whereas 12 miRNAs were highly (≤−5 fold) downregulated in high-fertile bull sperm and seminal plasma. It should be noted that highly differentially expressed (≤−5 and ≥5-fold regulation) miRNA were considered for further analysis and for the discussion. Table 3 shows top integrated genes of the DE-miRNAs predicted biological processes demonstrating their critical roles in the regulation of sperm function, fertilization, uterine receptivity, and in the development of embryo, placenta, and fetus. 

Seminal plasma has an enriched repertoire of seminal exosomes, which carry a unique profile of miRNAs [110]. A consensus is that seminal exosomes represent a mixed population of extracellular vesicles originating not only from the epididymis but also from accessory sex glands. Nevertheless, seminal exosomes mediators improve sperm motility and induce sperm to have the ability to capacitate and complete acrosomal exocytosis [111,112]. In addition, seminal plasma exosomes modulate the immune response and molecular signaling cascades in the uterus to facilitate embryo implantation [113,114] and these functions are mediated via seminal exosome-carried miRNAs [110].

Free and exosomal miRNAs in seminal plasma can reflect the pathophysiological conditions of the organ of origin. Aberrant cell-free miRNA levels, both in whole seminal plasma [20,109,115] and in seminal plasma exosomes [116] were linked to the poor quality of the sperm. Therefore, the analysis of cell-free miRNAs repertoire could be a specific indicator of alterations of the organ/cells of origin. In the current study, DE-miRNAs in sperm and seminal plasma are found to be common. Previously, we compared expression levels of individual miRNA both in sperm and seminal plasma from the same ejaculate [9,10]. The results revealed that the sperm miRNA level was significantly greater compared with the seminal plasma miRNA level and interestingly, the expressions of DE-miRNAs in sperm and seminal plasma followed similar trends for all miRNAs studied [10]. It also has been reported by others that approximately 75% of miRNAs in boar sperm and seminal plasma are shared [117]. Although a similar pattern of miRNAs’ expression between sperm and seminal plasma exists, the genetic and epigenetic pathways controlled by sperm and seminal plasma could be different. For example, exosomal miRNAs in the seminal plasma seem to not only interact with the spermatozoa but also with cells from the female reproductive tract, modulating their gene expression and influencing the female immune response triggered by the semen [23]. Further, sperm miRNAs are sensitive to environmental stimuli that can modify the sperm miRNA profile and might induce epigenetic modifications in the embryo [118].

The success of artificial insemination (AI) and in vivo and in vitro embryo transfer in cattle indicates that seminal plasma is not required for the establishment of pregnancy. Removal of seminal vesicles had no effect on the fertility of sperm [119]. Intrauterine infusion of 0.5 mL of seminal plasma at artificial insemination in dairy cows did not improve pregnancy/AI (saline, 52.4 (164/313) vs. seminal plasma, 45.5 (143/314) [120] and seminal plasma (37.8) vs. no treatment (33.2%)) [121]. It is plausible that the treatment regimen used in those studies did not cause the amount of inflammation typically induced by natural mating. It should be noted that, following natural mating, an unknown number of seminal molecules are being transferred to the uterus directly by the seminal plasma or carried by sperm [122]. Seminal plasma does not simply protect sperm but can also influence the reproductive events independent of sperm in some species other than cattle. Seminal fluid regulation of female reproductive physiology is shown in hamsters, pigs, and human studies [68,73,123,124,125,126,127,128,129,130,131,132,133,134]. Mating with males lacking seminal vesicles resulted in reduced blastocyst rate, reduced conception rates, and offspring that experienced sex-dependent phenotypic changes postnatally [120,125,135]. However, transformed offspring phenotype was ascribed to sperm damage and absence of seminal fluid [136]. It is evident that seminal plasma and sperm-borne mediators are critical for signaling events that lead to successful pregnancy and healthy offspring; however, there are differences across species [119,127,137] that raise the question of whether seminal plasma is a prerequisite in cattle breeding. It should be noted that the interaction of hub genes of DE-miRNAs in the PPI network was found in this current study.

Sellem et al. (2021) investigated the dynamics of bovine sperm small RNAs from spermatogenesis to final maturation and observed that sperm progressively acquires miRNAs during the epididymal transit with regional specificities [138]. Several miRNAs are involved in cell cycle regulation, spermatogenesis, or embryo development, suggesting the role of epididymal small RNAs in sperm maturation and for ejaculated sperm in embryogenesis. The GO annotation terms from in silico network analysis in this study predicted several functions pertinent to sperm and seminal plasma, and events relevant to fertilization and embryo and placental development. The nested network from in silico interactions revealed that upregulated miRNAs and their integrated genes were involved in cortical cytoskeleton organization, cyclin-dependent protein kinase holoenzyme complex structure, miRNA-mRNA interaction, cyclin-dependent protein serine/threonine kinase activity, phosphotyrosine residue binding, cancer biology, regulation and migration of T-cell, and cellular senescence signaling pathways.

Cytoskeleton organization has been implicated in placental development by regulation of ion transport in humans34, sperm-oocyte interaction [139], and regulation of cell architecture in mouse embryonic stem cells [140]. Actin filaments of the cytoskeleton participate in several vital phases of spermatogenesis including acrosome biogenesis, flagellum formation, nuclear processes modulating capacitation and acrosome reaction, DNA integrity, and fertility [141]. It is interesting to note that protein kinase c (PKC) has a role in the reorganization of the cortical cytoskeleton during the transition from oocyte to the fertilization-competent egg in response to progesterone [142]. Further, PKC-delta regulates bovine embryo development and gene expression in bovine trophoblast cells [143]. Serine/threonine regulation is important for phosphorylation associated with sperm capacitation [75]. Further, it is involved in placental development since the reduction of phosphorylation and endothelial nitric-oxide synthase may cause impaired placenta development, fetal growth retardation, and neonatal mortality [76].

Nested network in silico investigation, especially PANTHER GO prediction for DE miR-20b revealed that T cell migration and regulation are key functions component of uterine receptivity and embryo implantation. The role of seminal plasma in immune regulation of uterine receptivity has been studied in several species [70,124]. Regulatory T cells induced by seminal plasma assist embryo implantation by suppressing inflammation, inhibiting effector immunity towards the embryo, and promoting uterine vascular adaptations that support placental development [70,72,73]. Breeding by seminal plasma-deficient hamsters resulted in reduced fertility, impaired embryonic and fetal development, altered growth trajectories, and increased anxiety in offspring [68,123,125]. Different anti- and pro-tolerogenic factors such as cytokines, pregnancy hormones, seminal fluid, and decidua-specific cells, can modulate the number and the functions of regulatory T cells during pregnancy. Neutrophils and T cells facilitate maternal immune tolerance to paternal alloantigens, and neutrophils interact with T cells to promote angiogenesis at the maternal-fetal interface [72,73,124]. It should be noted that regulatory T cells promote the expansion of uterine mast cells and normalize early pregnancy angiogenesis in abortion-prone mice [26]. In pigs, miRNAs involved in immune cell development and possible regulation in abortion [144]. Significant differences were observed for miRNAs involved in immune cell development and angiogenesis (miR-296-5P, miR-150, miR-17P-5P, miR-18a, and miR-19a) between endometrial lymphocytes associated with healthy and arresting conceptus attachment sites. These pieces of evidence support the mediation of semen miRNAs in T-cell regulation and embryo implantation [72,73].

Protein tyrosine phosphorylation is involved in sperm-oocyte interaction, penetration, and fertilization [127]. Protein tyrosine phosphatase-interacting protein 51 (PTPIP51/RMD3) mRNA expression was seen in syncytiotrophoblast and cytotrophoblast layers of placentae and serves as cellular signaling partner in angiogenesis and vascular remodeling in women [128].

The nested network from in silico interaction analysis revealed that downregulated miRNAs and their predicted genes involved in advanced glycation end products (AGE)- receptor for AGE (RAGE) signaling, protease binding, regulation of actin cytoskeleton reorganization pathway, neurotrophin signaling, and histone phosphorylation pathways. Interfering with these pathways could impact reproductive success.

AGE/RAGE signaling has been shown to increase oxidative stress and inflammation. AGE impaired testicular function and thus affected spermatogenesis and sperm quality in rodents [134]. AGE hindered trophoblast invasion. Placental RAGE was activated during preeclampsia and caused inflammation in the trophoblast [130].

Neurotrophins and hormones influence each other and regulate neuropeptide expression. The PANTHER GO annotation for the DE-miRNAs and integrated genes envisaged neuropeptide and G protein-coupled receptor signaling. Ligands for G protein-coupled receptors (GPCR) are capable of activating mitogenic receptor tyrosine kinases (Trk) [131]. Neurotrophins utilize receptor Trk, namely TrkA, TrkB, and TrkC. Decreased Trk receptor activity by adenosine resulted in increased cell survival [132]. The cellular and molecular basis for the integration of metabolism and reproduction involves a complex interaction of hypothalamic neuropeptides with metabolic hormones (leptin, ghrelin, and IGF) and sex steroids. Neuropeptides including galanin-like peptide (GALP), neuropeptide Y (NPY), products of the proopiomelanocortin (POMC), and kisspeptin, serve as molecular motifs integrating metabolism and reproduction [137]. In heifers, increased rates of BW gain during a prepubertal period can advance puberty [145,146]. Functional changes in the NPY and POMC neuronal pathways appear to contribute to attaining puberty [134]. In humans, miR-21-3p has been implicated in adipose tissue browning and diabetes [147]. Sperm and seminal plasma adiponectins were associated with sperm parameters [148] and circulating adiponectin was linked to body condition, uterine inflammation, and sire conception rate [34,149]. G protein-coupled receptor is involved in spermatogenesis and sperm maturation [150], uterine receptivity [151], embryo implantation and development [152,153], and placental development [154]. 

Histone modification facilitates histone-to-protamine transition during spermiogenesis. It modulates chromatin compaction and higher-order chromatin structure by ubiquitination and methylation. Flaws in either the replacement or the alteration of histones might cause male infertility such as azoospermia, oligospermia, or teratozoospermia [155]. Disruption of histone variant H3f3a produces abnormal spermatozoa [156,157], and the loss of H3f3b leads to growth defects and death at birth, with surviving H3f3b-null males showing complete infertility [158]. Acrosome proteases trypsin and chymotrypsin play a role in acrosome reaction in bovine sperm. Sperm incubated with the inhibitors of the acrosome proteases reduced the percentage of acrosome reaction [159]. Serine protease PRSS55 affected sperm migration and sperm-egg binding in mice by interfering with ADAM3 [160].

The involvement of DE-miRNAs and their integrated genes in several reproductive functions in high-fertile sperm and seminal plasma in the current study, and various other studies, are presented in Table 3. It is not surprising that DE-miRNAs in the current study, regulate two events, focal adhesion, and cellular senescence. Focal adhesions are contact points that facilitate several cellular processes including proliferation, migration, and differentiation [35]. Several proteins including cadherins are involved in adhesion and signaling [161]. Cadherins are expressed in mature sperm and facilitate capacitation and sperm-oocyte interaction [162]. Cell aggregation, cell to cell contact formation, and cellular tension are important in embryonic development [163]. Programmed cellular senescence is critical to promote tissue remodeling during embryonic development. This process is regulated by the TGF-β/SMAD and PI3K/FOXO pathways [164], which are linked to DE-miRNAs from this study. 

Further, DICER1 and AGO2 among the predicted target genes for DE-miRNAs may notably have substantial influence on spermatogenesis and embryo development. Deletion of DICER gene in male germ cells led to impaired differentiation of haploid spermatids, followed by apoptosis and failure of spermatogenesis in the haploid and meiotic stages [165]. Though this is not a sperm/seminal plasma-induced reproductive event, increased expression of DICER1 and AGO2 were observed when porcine blastocysts underwent rapid and radical morphological changes from spherical, through tubular, to filamentous forms [166]. Several sequencing studies also revealed that DE-miRNAs, miR-34a, and miR-200a, inhibited endometrial receptivity and embryo implantation. MicroRNA-34a [167], and miR-200a [98], were downregulated in sperm and seminal plasma of high-fertile bulls in the current study.

Sperm morpho-functional characteristics are not always sufficient to predict male fertility potential. Sperm carry different molecules (proteins, RNAs, and ncRNAs), which are involved in important functions, such as spermatogenesis, sperm maturation, fertilization, and embryo development. The sperm acquire motility and oocyte-fertilizing ability during the epididymal passage. During their transit through the different epididymal segments, sperm come into contact with different repertoires of miRNAs, transcripts, and proteins, that are released from the epididymal epithelium mostly via epididymosomes [24,168,169]. Biogenesis of miRNAs seems to be important in the regulation of epididymal epithelium, sperm maturation, and fertility. While miR-10a/b, -21a, -29c, -196b-5p, -199a, -200b/c, and -208b-3p, accrue in sperm during passage through the epididymis; miR-204b-5p, and miR-375-3p, are more abundant in sperm from the caput and corpus of the epididymis [24,169]. In addition, epididymosomes also display different miRNA profiles and traffic small RNAs to sperm [24], including miR-143, -145, -199, and -214, that are more abundant in epididymosomes of the caput of the bovine epididymis, and miR-395, -654, and -1224, that are more abundant in epididymosomes from the cauda [169]. The miRNAs are acquired during the late steps of spermatogenesis or post-testicular sperm maturation with a potential function at the time of fertilization [24,170] (Sharma et al., 2016; Yuan et al., 2016).

Our previous studies found associations between fertility and sperm parameters. Sire conception rate showed a positive correlation with relative sperm volume shift [35] and acrosome reacted sperm at 5 h post-thaw [34], no correlation with hypoosmotic swelling [35], negative correlation with lipid peroxidation [171], sperm DNA fragmentation, and plasma membrane integrity [172]. Turrri et al. (2021), observed a positive association of field fertility (estimated relative conception rate), semen quality parameters (sperm kinetics and DNA), and specific miRNAs (15 differentially expressed miRNAs (including nine known miR-2285n, miR-378, miR-423-3p, miR-191, miR-2904, miR-378c, miR-431, miR-486, miR-2478)) between high- and low-fertility bulls [21]. 

In the current study, a bull field fertility parameter (sire conception rate) was utilized to distinguish high- and low-fertile bulls. We observed that 56 out of 84 (66.7%) miRNAs were expressed >2 fold and 32 out of 84 (38%) miRNAs were expressed >5 fold in sperm and seminal plasma by RT-PCR. Keles et al. (2021), used non return rate as fertility parameter and identified 85 differentially expressed sperm miRNAs by RNA sequencing [173]. The miR-2340, miR-26a, miR-425-5p, and miR-151–5p, were moderately correlated to nonreturn rate when conventional semen was used, whereas nine miRNAs were significantly correlated (miR-9-5p, miR-34c, miR-449a, miR-2483-5p, and miR-21–5p were negatively related, and miR-423-5p, miR-1246, miR-92a, and miR-5193-5p, were positively related) when gender selected semen was used. 

In the current study, 2 miRNAs (mir-16b and mir-29c) in sperm and 4 miRNAs (mir-16b, mir-29c, 200a, and mir-101) in seminal plasma were very highly down regulated (10 fold; *p* < 0.001); whereas 9 miRNAs (mir-215, mir-17-5p, mir-199a-3p, mir-193a-3p, mir-142-5p, mir-21-3p, mir-20b, mir-199a-5p, and mir-214) in sperm and 6 miRNAs (mir-193a-3p, mir-142-5p, mir-21-3p, mir-20b, mir-199a-5p, and mir-214) in seminal plasma were highly upregulated (10 fold; *p* < 0.001). Of these, miR-29c, miR-101, miR-200a, miR-20b, and miR-214, were associated with hub genes and these miRNA-mRNA pairs were involved in spermatogenesis, embryogenesis, placenta development, and embryo implantation. The cluster analysis revealed their involvement in the regulation of spermatogenesis, histone modifications, ovarian follicle development, focal adhesion, cellular senescence, vascular remodeling and placental development, and VEGF and AGE-RAGE signaling.

Previous studies observed positive (upregulated-upregulated) or negative (upregulated-downregulated) association between miRNA and mRNA pairs [174]. The miRNAs are capable of activating gene expression directly or indirectly in response to different cell types and environments and in the presence of distinct cofactors. The biological outcome of miRNA-mRNA interaction can be altered by several factors contributing to the binding strength and repressive effect of a potential target site. Interestingly, miRNA-mRNA associations were complementing in normal tissues and miRNA-mRNA associations were reverse complementing in the same tissues in diseased condition [175]. The correlation between miRNA and hub gene expressions were negative in the current study.

Studies have shown that interaction between seminal plasma and sperm could improve acrosome integrity, sperm quality by modulating sperm potential to evoke in vitro capacitation, and cause progesterone-induced acrosomal exocytosis [176]. During sperm preservation, the impact of seminal plasma on sperm has been unclear. Seminal plasma is diluted/removed prior to cryopreservation to reduce the association between seminal plasma and sperm during storage. An early study revealed that top 10 highly expressed miRNA predicted target genes (across the three libraries such as ejaculated sperm, epididymal sperm, and seminal plasma) could be involved in spermatogenesis, zygote formation, and animal-environment interactions [117]. This shows the importance of contact between sperm and seminal plasma before preservation [177].

There are limitations to this study. First, the study is limited due to small sample size. Though four bulls per group was calculated as adequate to determine fold regulation differences of five, it should be noted that the small number of bulls were represented from general population. We anticipate that more miRNA and genes can be identified in future studies with a large sample size. Second, the identified miRNA-mRNA interactions were mostly based on predictions from public databases. The causal regulations of each pair and the underlying mechanisms require further functional characterization in future studies.

## 5. Conclusions

In conclusion, we observed miRNAs were differentially expressed in sperm and seminal plasma of high-fertile bulls compared with low-fertile bulls. The DE-miRNAs fold changes were similar in both sperm and seminal plasma. The DE-miRNAs in sperm and seminal plasma elucidated in this study plausibly regulated critical pathways that are required for reproductive success at the transcriptional or post-transcriptional level. In silico network analysis showed association for the tasks of sperm and seminal plasma DE-miRNAs and its predicted genes specific to spermatogenesis, sperm maturation, sperm and seminal plasma interaction, early embryo development, implantation, and organogenesis.

## Figures and Tables

**Figure 1 animals-12-02360-f001:**
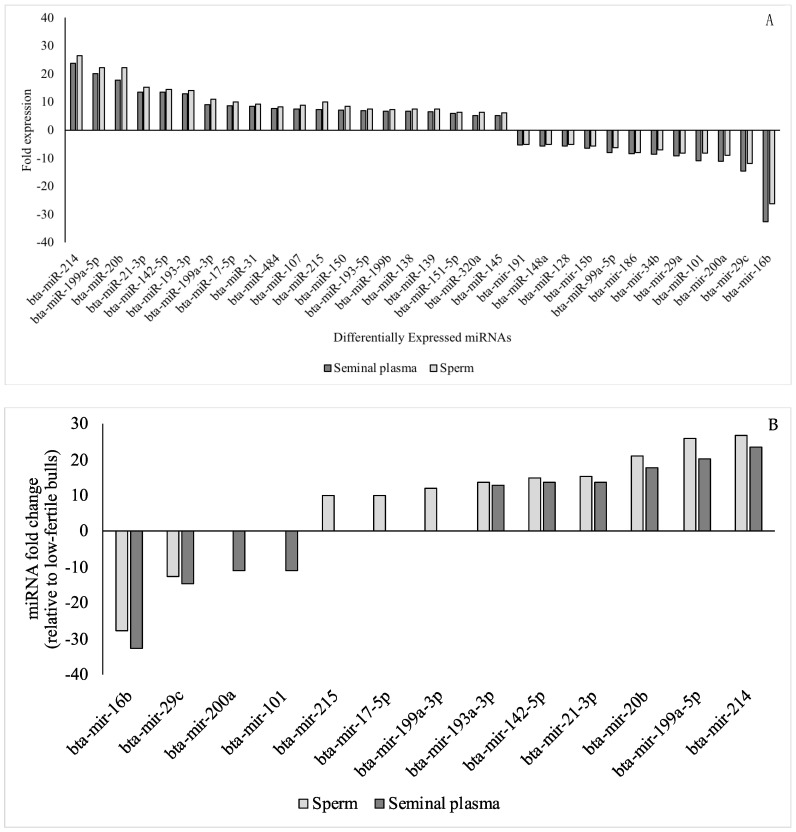
(**A**) Fold regulation of differentially expressed sperm and seminal plasma miRNAs in high-fertile compared to low-fertile Holstein bulls. Of 84 bovine-specific well-characterized miRNAs investigated, 20 were greater (*p* ≤ 0.05; fold ≥ 5) and 12 were lower (*p* ≤ 0.05; fold ≤ −5) in sperm and seminal plasma in high-fertile Holstein bulls; (**B**) Fold regulation of highly differentially expressed sperm and seminal plasma miRNAs in high-fertile compared to low-fertile Holstein bulls. Of 84 bovine-specific well-characterized miRNAs investigated, 11 miRNA [9 upregulated (≥10) and 2 down regulated (≤−10)) were differentially expressed in high compared with low-fertile sperm (*p* < 0.001); whereas 10 miRNA (6 upregulated (≥10) and 4 down regulated (≤−10)) were differentially expressed in high compared with low-fertile seminal plasma.

**Figure 2 animals-12-02360-f002:**
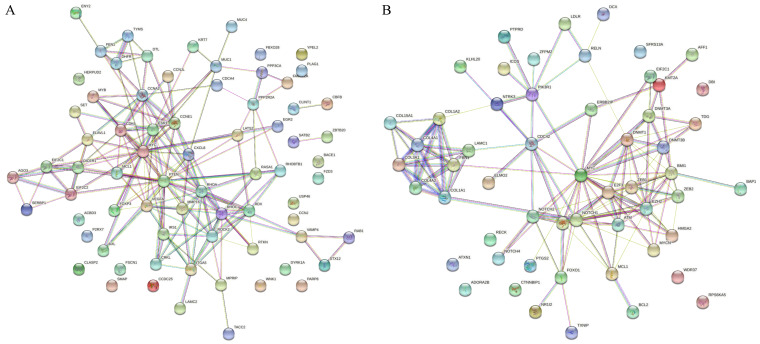
STRING protein-protein interaction (PPI) network. (**A**) PPI network for the upregulated DE-miRNAs predicted 75 genes (73 nodes and 188 edges, PPI enrichment *p* < 1.0 × 10^−16^; (**B**) PPI for the upregulated DE-miRNAs predicted 57 genes (56 nodes and 186 edges, PPI enrichment *p* < 1.0 × 10^−16^); the color nodes represent proteins. The edges represent interactions.

**Figure 3 animals-12-02360-f003:**
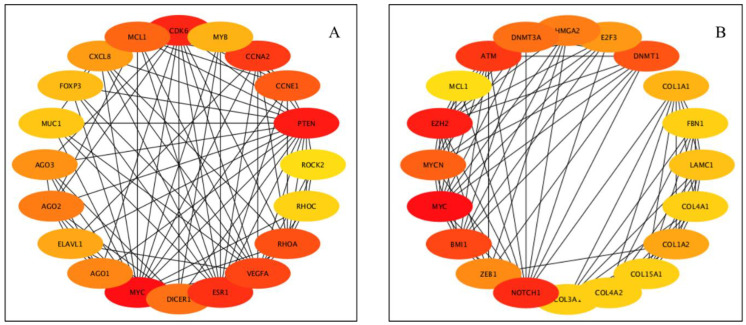
Interaction of hub genes of DE-miRNAs in the PPI network. (**A**) PPI network of top genes for highly upregulated DE-miRNAs; (**B**) PPI network of the top genes for highly downregulated DE-miRNAs. DE-miRNAs, differentially expressed microRNAs; PPI, protein-protein interaction; the PPI among hub genes for upregulated DE-miRNAs were greater compared with hub genes for down-regulated DE-miRNAs. Color red to yellow denotes high to low degree of expression. Black lines indicate interactions between genes.

**Figure 4 animals-12-02360-f004:**
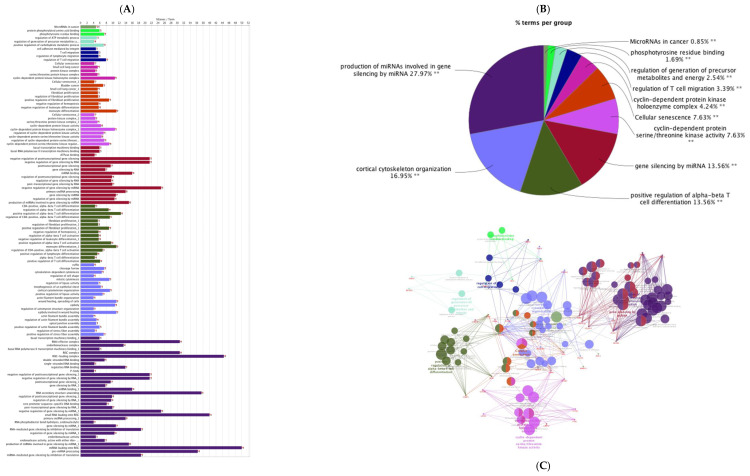
ClueGO analysis of up-regulated genes in sperm and seminal plasma from high-fertile bulls. (**A**) GO/pathway terms specific for upregulated genes. The bars represent the number of genes associated with the terms. The percentage of genes per term is shown as bar label; (**B**) overview chart with functional groups including specific terms for upregulated genes; (**C**) functionally grouped network with terms as nodes linked based on their kappa score level (≥0.4), where only the label of the most significant term per group is shown. The node size represents the term enrichment significance. Functionally related groups partially overlap. The color gradient shows the gene proportion of each cluster associated with the term. Single (*) or double (**) asterisk indicate significant enriched GO terms at the *p* < 0.05 and *p* < 0.01 statistical levels.

**Figure 5 animals-12-02360-f005:**
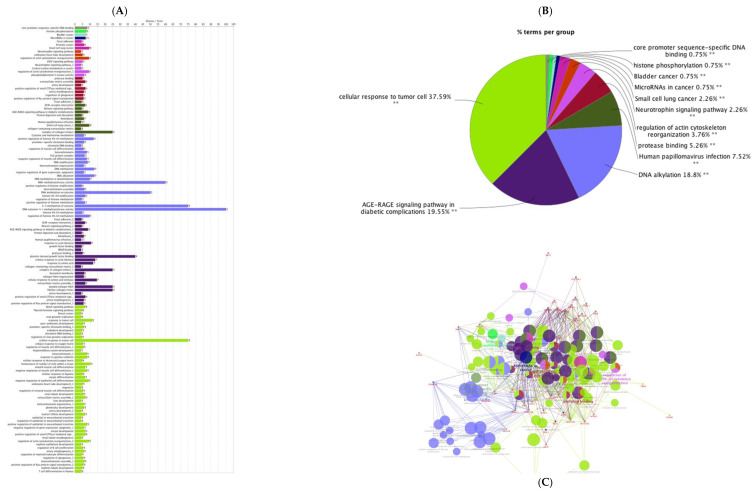
ClueGO analysis of down regulated genes in sperm and seminal plasma from high-fertile bulls. (**A**) GO/pathway terms specific for down regulated genes. The bars represent the number of genes associated with the terms. The percentage of genes per term is shown as bar label; (**B**) overview chart with functional groups including specific terms for down regulated genes; (**C**) functionally grouped network with terms as nodes linked based on their kappa score level (≥0.4), where only the label of the most significant term per group is shown. The node size represents the term enrichment significance. Functionally related groups partially overlap. The color gradient shows the gene proportion of each cluster associated with the term. Single (*) or double (**) asterisk indicate significant enriched GO terms at the *p* < 0.05 and *p* < 0.01 statistical levels.

**Figure 6 animals-12-02360-f006:**
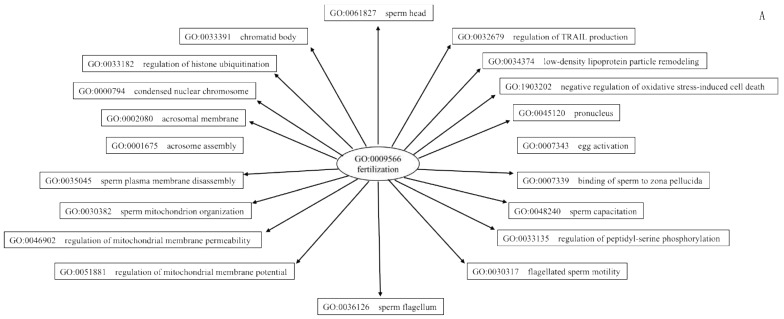
(**A**) Selected QuickGo gene ontology term and its co-occurring terms predicting processes involved fertilization. Note: GO terms male organ development and fertilization predicted from Database for Annotation, Visualization and Integrated Discovery (DAVID 6.8) bioinformatics analysis for differentially expressed miRNAs both in high-fertile sperm and seminal plasma were used for prediction of co-occurring QuickGO terms. (**B**) Selected QuickGo gene ontology term and its co-occurring terms predicting processes involved progeny development. Note: GO terms embryo, trophoblast, and organ development predicted from Database for Annotation, Visualization and Integrated Discovery (DAVID 6.8) bioinformatics analysis for differentially expressed miRNAs both in high-fertile sperm and seminal plasma were used for prediction of co-occurring QuickGO terms.

**Figure 7 animals-12-02360-f007:**
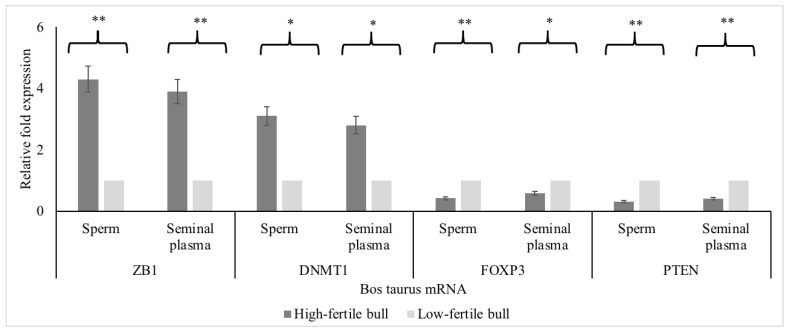
mRNA expression of hub genes in sperm and seminal plasma of high- and low-fertile bulls. *DNMT1*, DNA methyltransferase 1; *FOXP3*, forkhead box P3 (scurfin); *PTEN*, phosphatase, and tensin homolog; *ZEB1*, Zinc Finger E-Box Binding Homeobox 1; *GADPH*, glyceraldehyde-3-phosphate dehydrogenase. Single (*) or double (**) asterisk indicate significant enriched GO terms at the *p* < 0.05 and *p* < 0.01 statistical levels.

**Figure 8 animals-12-02360-f008:**
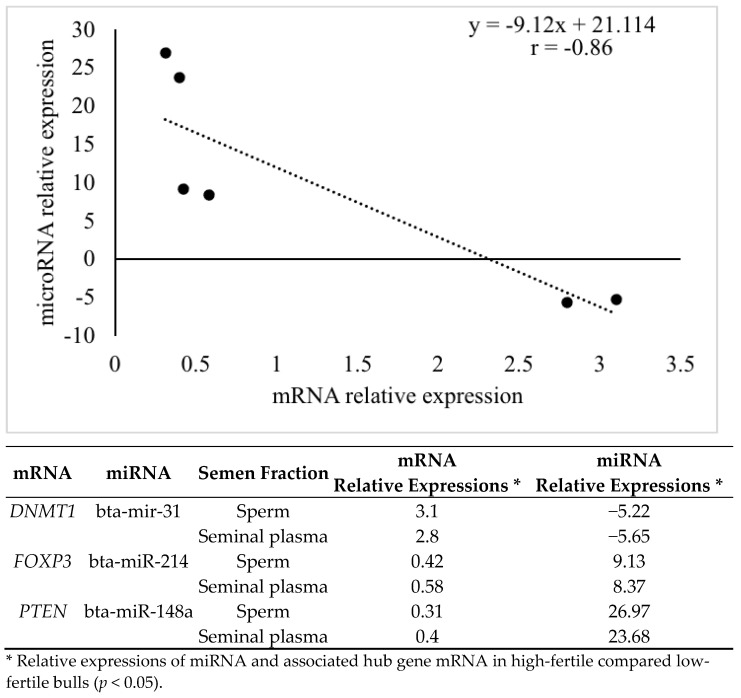
Correlation coefficient (*r*) of mean relative expressions of miRNA and mRNA pairs. Pearson correlation coefficient *p* < 0.05.

**Table 1 animals-12-02360-t001:** Bovine miRBase profiler plate, consisting of primers for 84 target miRNAs and control genes.

Layout	1	2	3	4	5	6	7	8	9	10	11	12
A	bta-let-7f	bta-miR-101	bta-miR-103	bta-miR-125a	bta-miR-125b	bta-miR-126-3p	bta-miR-128	bta-miR-145	bta-miR-148a	bta-miR-151-3p	bta-miR-151-5p	bta-miR-16b
B	bta-miR-181a	bta-miR-18a	bta-miR-18b	bta-miR-199a-5p	bta-miR-205	bta-miR-20a	bta-miR-21-5p	bta-miR-221	bta-miR-222	bta-miR-26a	bta-miR-26b	bta-miR-27a-3p
C	bta-miR-27b	bta-miR-29a	bta-miR-300-5p	bta-miR-30d	bta-miR-31	bta-miR-320a	bta-miR-34b	bta-miR-484	bta-miR-499	bta-miR-99a-5p	bta-miR-7a-5p	bta-let-7d
D	bta-let-7g	bta-let-7i	bta-miR-17-5p	bta-miR-107	bta-miR-10a	bta-miR-10b	bta-miR-122	bta-miR-124b	bta-miR-127	bta-miR-132	bta-miR-138	bta-miR-139
E	bta-miR-140	bta-miR-142-3p	bta-miR-142-5p	bta-miR-148b	bta-miR-150	bta-miR-15b	bta-miR-17-3p	bta-miR-17-5p	bta-miR-181b	bta-miR-181c	bta-miR-186	bta-miR-191
F	bta-miR-192	bta-miR-193a-3p	bta-miR-193a-5p	bta-miR-199a-3p	bta-miR-199b	bta-miR-200a	bta-miR-200b	bta-miR-200c	bta-miR-20b	bta-miR-210	bta-miR-21-3p	bta-miR-214
G	bta-miR-215	bta-miR-218	bta-miR-22-5p	bta-miR-23a	bta-miR-23b-3p	bta-miR-24-3p	bta-miR-25	bta-miR-29b	bta-miR-29c	bta-miR-30a-5p	bta-miR-30c	bta-miR-30e-5p
H	cel-miR39-3p	cel-miR39-3p	SNORD42B	SNORD69	SNORD61	SNORD68	SNORD96A	RNU6-6P	miRTC	miRTC	PPC	PPC

Characterized 84 target miRNAs (plate well positions A1–G12) and controls (plate well positions H1–H12).

**Table 2 animals-12-02360-t002:** Forward and reverse primer sequence for quantitative real-time polymerase chain reaction amplification of mRNA for canine testis samples.

Gene	Forward Primer	Reverse Primer	Product Length	Accession Number
*ZEB1*	AAAGCAGCAGGGCGAGTTAT	TATGGGGTTGGCACTTGGTG	181	NM_001206590.1
*DNMT1*	TATCGGCTGTTCGGCAACAT	GGCAGCCTCCTCCTTGATTT	153	NM_182651.2
*FOXP3*	CAGCGGACACTCAACGAGAT	AACTCATCCACGGTCCACAC	164	XM_024987818.1
*PTEN*	GCAGCTTCTGCCATCTCTCT	ATGCTTTGAATCCAAAAACCTTACT	235	NM_001319898.1
*GADPH*	GTGAAGGTCGGAGTGAACGG	ATTGATGGCGACGATGTCCA	93	NM_001034034.2

*DNMT1*, dna methyltransferase 1; *FOXP3*, forkhead box P3 (scurfin); *PTEN*, phosphatase and tensin homolog; *ZEB1*, Zinc Finger E-Box Binding Homeobox 1; *GADPH*, glyceraldehyde-3-phosphate dehydrogenase.

**Table 3 animals-12-02360-t003:** Differentially expressed miRNAs, associated predicted hub genes and their linked reproductive functions.

Upregulated miRNAs	Hub Genes	Reproductive Functions	Species	References
miR-107	*AGO1, AGO2, AGO3, CCNE1, CDK6*	Sperm function	Bovine & human	[37,38,39]
		Angiogenesis	Human	[40]
		Embryo development	Porcine & invertebrates	[41,42,43]
		Placental development	Human	[44]
miR-132	*RHOC*	Sperm maturation, sperm parameters	Human & murine	[45,46,47,48]
		Trophoblast development	Human	[25]
		Embryo development	Murine	[49]
miR-138	*ROCK2*	Placental development	Human	[46,50]
		Embryo development	Swine	[51]
		Embryo-placenta interaction	Murine	[52]
		Organogenesis	Human	[53]
miR-145	*CCNA2, MYC, MUC1, CXCL8*	Spermatogenesis and function	Human	[54,55]
		Embryo development	Bovine & murine	[24,56]
		Trophoblast development	Human	[57]
		Embryo implantation	Bovine & Murine	[58,59,60]
miR-150	*CCNE1, MYB*	Spermatogenesis	Human	[61]
		Embryo development	Human	[62]
		Trophoblast development	Human	[63]
miR-20b	*VEGFA, ESR1*	Spermatogenesis	Buffalo	[64]
		Pregnancy establishment	Bovine	[65,66]
		Trophoblast development	Human	[67]
		Immune response, uterine receptivity	Swine, murine & hamster	[68,69,70,71,72,73,74,75]
miR-214	*PTEN*	Spermatogenesis, sperm DNA	Canine & murine	[76]
		Embryo development	Human	[69]
		Uterine capacity and liter size	Swine	[77]
miR-31	*RHOA, ELAVL1, FOXP3, DICER1*	Sperm DNA	Human	[78]
		Fertility, embryo development	Human, mouse, rat & invertebrates	[79]
		Embryo implantation	Human	[80,81]
		Organogenesis	Human, mouse, rat & invertebrates	[79]
miR-320a	*MCL*	Embryo development	Human, bovine & rat	[17,82,83,84]
miR-101	*MYCN, MCL1, ATM, EZH2*	Sperm parameters	Human	[85]
		Placental development	Human	[86]
		Embryo implantation	Murine	[87,88]
miR-128	*BMI1, E2F3*	Spermatogenesis, sperm maturation	Murine	[86]
		Angiogenesis	Human & rat	[89,90]
		Endometrium programming	Bovine	[91]
		Trophoblast development	Human	[17]
miR-148a	*DNMT1*	Spermatogenesis	Human	[92]
		Angiogenesis	Human	[93]
		Placenta development	Human & rat	[94,95]
		Embryo development	Swine	[78] (Weng, Peng)
miR-200a	*ZB1*	Spermatogenesis	Canine, Murine	[76,96]
		Embryo development	Murine	[97]
		Embryo implantation/Uterine receptivity	Murine	[98,99]
		Trophoblast development	Murine	[100]
miR-29c	*COL1A1, COL1A2, COL4A2, FBN1, LAMC1, COL4A1, COL3A1, COL15A1, DNMT3A*	Sperm DNA	Human	[101]
		Endometrium programming	Human	[102,103]
		Embryo implantation	Human	[103,104]
miR-34b	*NOCH1, MYC, HMGA2*	Spermatogenesis	Murine	[105,106,107]
		Embryo implantation	Human	[108]
		Fetal development	Murine	[109]

## Data Availability

The original contributions presented in the study are included in the article/Appendix A; further inquiries can be directed to the corresponding author.

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
