# Peer review of "Investigation of Sperm and Seminal Plasma Candidate MicroRNAs of Bulls with Differing Fertility and In Silico Prediction of miRNA-mRNA Interaction Network of Reproductive Function"

_animals, 2022, doi:10.3390/ani12182360_

Round 1

Reviewer 1 Report

The research team identified the gene as a candidate for reproductive function by discovering the bull's microRNA candidate and confirming the in silico prediction of the miRNA-mRNA interaction network. Through the miRNA-mRNA interaction network, the relationship between useful genes has been identified and is expected to be useful in the future.

It is a pity that this paper did not use these prediction results.

They check the gene expression level in bull's sperm and plasma, and select genes with large differences in expression.

And, if the gene had been deleted and the abnormal effects of sperm were confirmed, it would have been supported as a more convincing result.

I suggest that you explain the specific method of investigating progressive motility and abnormal spermatozoa (2-1).

According to the paper, it is expected to be able to confirm that it is a highly capable sperm by investigating seminal plasma. I wonder why you didn't investigate the embryo development of in vitro fertilization.

Author Response

The research team identified the gene as a candidate for reproductive function by discovering the bull's microRNA candidate and confirming the in silico prediction of the miRNA-mRNA interaction network. Through the miRNA-mRNA interaction network, the relationship between useful genes has been identified and is expected to be useful in the future.

Authors: Thank you for the constructive review.

It is a pity that this paper did not use these prediction results. They check the gene expression level in bull's sperm and plasma and select genes with large differences in expression. And, if the gene had been deleted and the abnormal effects of sperm were confirmed, it would have been supported as a more convincing result.

Authors: We appreciate the comment. The deletion, and in vitro studies are on-going. Inclusion of those information in this current manuscript would stray from the current objectives and result in a long article.

I suggest that you explain the specific method of investigating progressive motility and abnormal spermatozoa (2-1).

Authors: Explanations are included as suggested (lines 100 and subsections 2.2.1 and 2.2.2)

According to the paper, it is expected to be able to confirm that it is a highly capable sperm by investigating seminal plasma. I wonder why you didn't investigate the embryo development of in vitro fertilization.

Authors: We thank the reviewer for the suggestion. The in vitro and in vivo embryo development investigation is ongoing. We will be submitting manuscripts in the recent future. Inclusion of those information in this current manuscript would stray from the current objectives and a long article.

Reviewer 2 Report

The authors elucidated that differentially expressed miRNAs in sperm and seminal plasma of high- and-low fertile bulls, integrate miRNAs to their target genes, and categorize target genes to predict biological processes in MS. The reviewer judged the MS is valuable for the publication in ANIMAL. However, the reviewer would be grateful if the authors could revise the following points.

General comments

1)    In general, cattle reproduction is mainly performed through AI with frozen/thawed sperm in cattle industry. In addition, in the process of frozen sperm, ejaculated/collected sperm is immediately separated from the seminal plasma by centrifuging after collection. And that ejaculated sperm is stored in the epididymides just before ejaculation, at ejaculation the sperm is contact with seminal plasma at once. Thereafter, the reviewer thought that the authors had better explain the relation between sperm and seminal plasm in cattle.

2)    The reviewer thought that the authors had better add explanations about genetic and phenotypic expression of the bulls selected as “high fertility” or “low fertility”, respectively.

3)    The reviewer thought that the MS is long, The reviewer would be grateful if the authors could rewrite “M & M” and Discussion.

Minor comments

-L111, 149, 252 : “Briefly” to “In brief” to avoid misunderstanding for readers.

Author Response

The authors elucidated that differentially expressed miRNAs in sperm and seminal plasma of high- and-low fertile bulls, integrate miRNAs to their target genes, and categorize target genes to predict biological processes in MS. The reviewer judged the MS is valuable for the publication in ANIMAL. However, the reviewer would be grateful if the authors could revise the following points.

Authors: Thank you for the constructive review.

General comments

In general, cattle reproduction is mainly performed through AI with frozen/thawed sperm in cattle industry. In addition, in the process of frozen sperm, ejaculated/collected sperm is immediately separated from the seminal plasma by centrifuging after collection. And that ejaculated sperm is stored in the epididymides just before ejaculation, at ejaculation the sperm is contact with seminal plasma at once. Thereafter, the reviewer thought that the authors had better explain the relation between sperm and seminal plasm in cattle.

Authors: The miRNAs transmigrate between sperm and seminal plasma. In the ejaculated sperm we observed the miRNA expression is greater in sperm compared with seminal plasma in boar semen. This information is included in the introduction, Lines 70  to 71. Our current study results showed that it is consistent with the findings from boar study.

Thus seminal plasma composition/contribution and contact between them before cryopreservation, is critical for sperm functional parameters - acrosome reaction, capacitation - that occur in the female reproductive tract, fertilization, and embryonic development. The discussion related to this was included (Lines 653-662).

2)    The reviewer thought that the authors had better add explanations about genetic and phenotypic expression of the bulls selected as “high fertility” or “low fertility”, respectively.

Authors: It was included from Lines 91 to 98 and in the Table S1.

3)    The reviewer thought that the MS is long, The reviewer would be grateful if the authors could rewrite “M & M” and Discussion.

Authors: We appreciate reviewer comments. We were aware of the length of the manuscript. However, in our humble opinion M & M were explained in a fashion that it would facilitate if someone wanted to repeat the study. So it is critical to balance the information. For an example the other reviewer requested to add assessment methods for progressive motility and abnormal spermatozoa to M&M.

We included reasonable and relevant biological explanation and we strongly feel that without it the discussion would be choppy.

Minor comments

-L111, 149, 252 : “Briefly” to “In brief” to avoid misunderstanding for readers.

Authors: It was corrected as suggested (Lines 126, 164 and 254)

Round 2

Reviewer 1 Report

I accept your opinion. It seems to be well organized with the opinions of all reviewers.